# Cost-effectiveness analysis of dapagliflozin for people with chronic kidney disease in Malaysia

**Soo Kun Lim**[1], **Shaun Wen Huey Lee**[2]*

1 Renal Division, Department of Medicine, Faculty of Medicine, University of Malaya, Kuala Lumpur, Malaysia, 2 School of Pharmacy, Monash University Malaysia, Subang Jaya, Selangor, Malaysia

* shaun.lee@monash.edu

## Abstract

### Introduction

Chronic kidney disease (CKD) is a global health concern which results in significant economic burden. Despite this, treatment options are limited. Recently, dapagliflozin has been reported have benefits in people with CKD. This study aimed to evaluate the cost–effectiveness of dapagliflozin as an add-on to standard of care (SoC) in people with CKD in Malaysia.

### Methods

A Markov model was adapted to estimate the economic and clinical benefits of dapagliflozin in people with Stage 2 to 5 CKD. The cost-effectiveness was performed based upon data from the Dapagliflozin and Prevention of Adverse Outcomes in Chronic Kidney Disease (DAPA-CKD) trial supplemented with local costs and utility data whenever possible.

### Results

In Malaysia, dapagliflozin in combination with SoC was the dominant intervention compared to SoC alone (RM 81,814 versus RM 85,464; USD19,762 vs USD20,644). Adding dapagliflozin to SoC in people with CKD increased life expectancy by 0.46 years and increased quality-adjusted life years (QALY) by 0.41 in comparison with SoC alone (10.01 vs. 9.55 years, 8.76 vs. 8.35 QALYs). This translates to a saving of RM8,894 (USD2,148) with every QALY gained. The benefits were due to the delay in CKD progression, resulting in lower costs of dialysis and renal transplantation. Results were robust to variations in assumptions over disease management costs as well as subgroup of population that would be treated and below the accepted willingness-to-pay thresholds of RM 46,000/QALY.

### Conclusion

The use of dapagliflozin was projected to improved life expectancy and quality of life among people with CKD, with a saving RM8,894 (USD2,148) for every quality-adjusted life-year gained and RM7,898 (USD1,908) saving for every life year gained.

**Data Availability Statement:** All relevant data are within the paper and its Supporting Information files.

**Funding:** This study is funded by AstraZeneca Malaysia. The funder had no role in study design,

data collection and analysis, decision to publish, or preparation of the manuscript.

**Competing interests:** The authors have declared that no competing interests exist.

## Introduction

Chronic kidney disease (CKD) is estimated to affect 8–16% people globally [1, 2]. In people with CKD, they are associated with an increased risk of all-cause and cardiovascular mortality and fractures [3–5]. This has a far reaching impact, and is associated health-related quality of life (HRQoL) impairments, but also has substantial societal and economic impact [6]. As such, there is significant benefits from halting or delaying CKD progression in the general population [7–9].

In Malaysia, the prevalence of CKD has been increasing over the past decade. The National Health and Morbidity Survey (NHMS) in 2011 reported that the prevalence of CKD was 9.07% [10]. This prevalence increased to 15.48% in the recent prevalence study in 2018 [11]. Despite the widespread problem of CKD, treatment options are limited; with only angiotensin-converting enzyme inhibitors (ACEi) and angiotensin receptor blockers (ARB) being the only medications that have consistently shown to slow disease progression [12–14]. Recently, sodium-glucose co-transporter-2 (SGLT2) inhibitor were reported to have cardiovascular and renal benefits in addition to improved glycaemic control [15, 16].

The efficacy and safety of dapagliflozin in addition to standard of care (SoC) in people with or without type 2 diabetes (T2DM) with CKD was investigated in the Dapagliflozin and Prevention of Adverse Outcomes in CKD (DAPA-CKD) study [17]. Patients on dapagliflozin were found to have a 39% lower risk of the composite primary endpoint of a $\geq$50% sustained decline in eGFR, onset of kidney failure, or incidence of cardiovascular or kidney-related death (Hazard ratio: 0.61; 95% confidence interval: 0.51 to 0.72; p<0.001).

Given these benefits, there is a need to assess the value of introducing dapagliflozin into a Malaysian national health formulary. Cost-effectiveness analyses represent an important tool to support informed decision making by policy makers. In this study, we assessed the cost effectiveness of the introduction of dapagliflozin in addition to SoC versus SoC in people with CKD from a Malaysian perspective.

## Method

### Model description

The present model is an adaptation based on the DAPA-CKD cost-effectiveness model which has been published previously [18] and implemented in Microsoft Excel 2019 (Richmond, VA). The model adopted a Malaysian health care perspective with a lifetime analytical horizon. Future cost and QALY were discounted 3% annually [19].

### Decision problem approach

The aim of the analysis was to investigate the cost-effectiveness outcomes in the general population with CKD [17], and thus spans a broad range of people with different estimated glomerular filtration rate (eGFR), urine albumin-creatinine ratio (UACR) $\geq$200 to $\leq$5000 mg/g and those with or without T2DM. The intervention and comparator are aligned to those of the informing trial data, where dapagliflozin 10mg was added to SoC compared to placebo in addition to SoC, respectively. The outcomes tracked in the analysis were mortality (all-cause and cardiovascular disease-specific) and CKD progression. The base case analysis was performed in Malaysia, with relevant utility tariffs, and costs applied where available. These individual outcomes are used to estimate the treatment effect on life years, quality-adjusted life years (QALY) and costs.

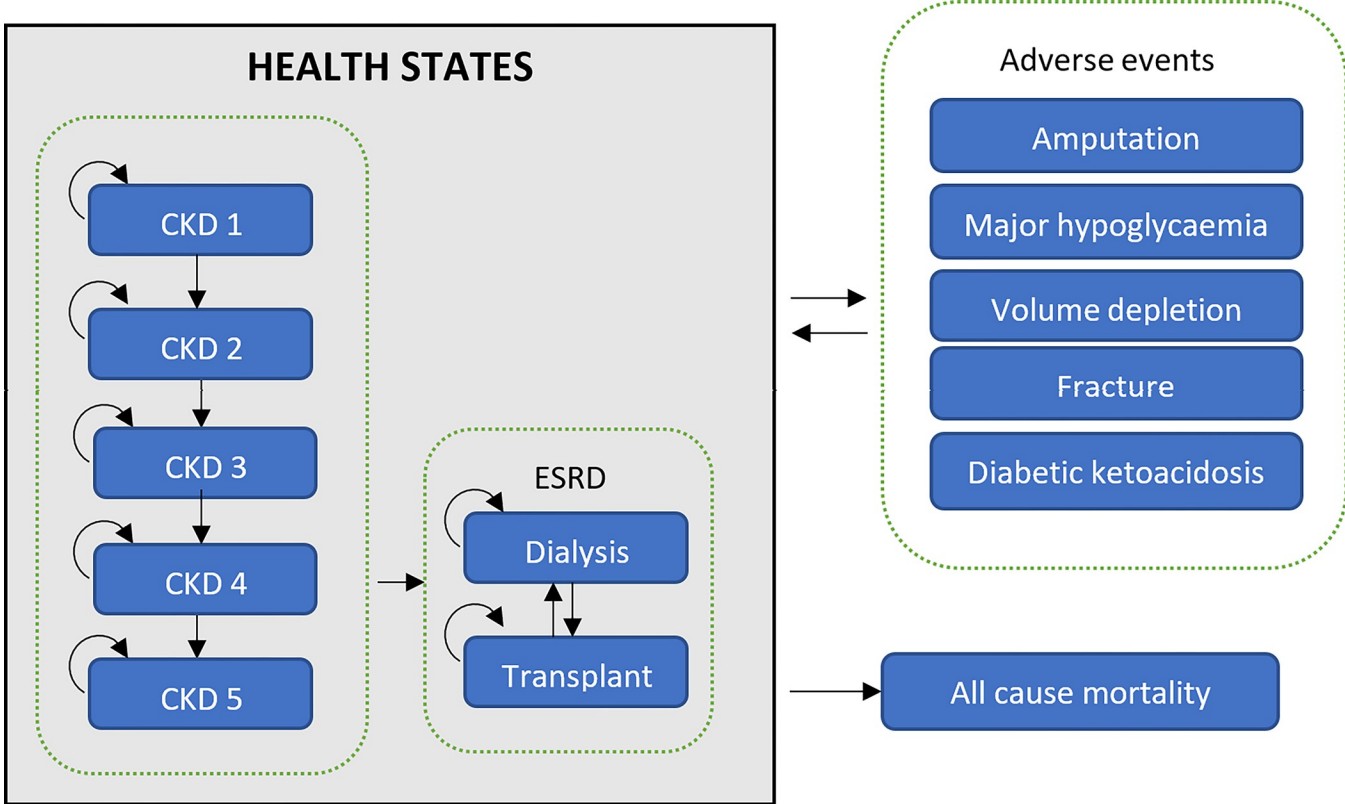

**Fig 1. Markov model diagram of this study.** Patients can begin at either of the eGFR health state prior to kidney failure. Once a patient experiences end stage kidney disease (ESRD), they can experience dialysis or transplant. Patients can suffer transient adverse events, incurring associated costs and disutility in the cycle of incidence. Death is end point.

## Model structure

The current model utilises a lifetime Markov state-transition framework with a 1-month cycle [20]. Disease progression was modelled through transitions between discrete health states characterised by CKD stage (defined by eGFR clinical laboratory values) and renal replacement therapy with state-specific utility decrements and outcomes. Health states describing these events were considered transient, i.e., patients remain in the heath state for one cycle, where they incur additional event specific costs and utility decrements. All patients could get worse and move to a more advanced CKD stage over time until they would require renal replacement therapy or could discontinue treatment due to other reasons. In the event a patient discontinue treatment, they are assumed to have the same transition and costs as patients receiving placebo (Fig 1).

## Model input

**Disease progression and mortality.** Disease progression was captured using parameters based upon the DAPA-CKD trial [17] data supplemented with locally relevant information where applicable [21]. For each arm, a transition matrix describing the movement between CKD health states were derived based on patient level data of the DAPA-CKD study [17] (S1 and S2 Tables). These are separated for the first 4 months of follow-up and from month 5 onwards to account for the differences in decline in eGFR observed initially. The model also captured the incidence of all-cause mortality. A constant rate of discontinuation was used.

**Table 1. Cost inputs used in the current study.**

| | Used cost, RM* (USD) | SE, RM (USD) | Reference |
|---|---|---|---|
| **Annual drug acquisition costs** | | | |
| Dapagliflozin | 463.55 (111.97) | 0.01 (0.00) | IMS Market Intelligence |
| SoC | 46.32 (11.19) | 0.01 (0.00) | |
| **Annual disease state management costs per annum+** | | | |
| CKD 1, >90ml/min per 1.73 m$^2$ | 343.91 (83.07) | 46.00 (11.11) | Calculated from micro-costing of study by Azmi [25] and Saminathan [11] |
| CKD 2, 60–89 ml/min per 1.73 m$^2$ | 484.07 (116.93) | 29.52 (7.13) | |
| CKD 3a, 45–59 ml/min per 1.73 m$^2$ | 425.63 (102.81) | 54.45 (13.15) | |
| CKD 3b, 30–44 ml/min per 1.73 m$^2$ | 1,287.52 (310.99) | 82.39 (19.90) | |
| CKD 4, 15–29 ml/min per 1.73 m$^2$ | 1,726.12 (416.94) | 27.84 (6.72) | |
| CKD 5 (pre-RRT); <15 ml/min per 1.73 m$^2$ | 3,194.31 (771.57) | 11,909.60 (2876.71) | |
| Dialysis | 44,731.40 (10,804.69) | 1,118.29 (270.12) | Surendra et al. [26] |
| Transplant (initial cost) | 117,805.11 (28,455.34) | 1,982.69 (478.91) | Bavanandan et al. [27] |
| Transplant (maintenance cost) | 25,281.64 (6106.68) | 632.04 (152.67) | |
| **Adverse event costs** | | | |
| Volume depletion | 1,330.90 (321.47) | 350.32 (84.62) | Pooled estimate of outpatient, inpatient data from 2 tertiary hospitals |
| Severe hypoglycaemic events | 8,802.57 (2126.22) | 462.78 (111.78) | Aljunid et al. [28] |
| Fractures | 7,862.22 (1890.39) | 391.31 (94.52) | Choo et al. [29] |
| Diabetic ketoacidosis | 8,802.57 (2126.22) | 462.78 (111.78) | Assumed same as severe hypoglycaemia |
| Amputation | 6,868.35 (1659.02) | 459.94 (110.10) | Feisul et al. [30] |

*inflated to 2022 value. Volume depletion was defined as an event that resulted in loss of fluid such as dehydration, hypovolemia, or hypotension requiring hospitalisation

**Treatment-related adverse events.** Dapagliflozin, like many other SGLT-2 inhibitors has a well-established safety profile. As such, in the current model, only grade 3 to 5 adverse events that have been found to be significantly different in the DAPA-CKD study [17] were accounted for, including volume depletion, fractures, diabetic ketoacidosis, severe hypoglycaemic events and amputation. The generalised estimating equation using Poisson distribution was used to estimate the recurrent events including volume depletion, fractures, diabetic ketoacidosis, severe hypoglycaemia and amputations.

**Health-related quality of life utilities and costs.** In this analysis, the utility inputs for patients CKD 2–5 were derived from the DAPA-CKD [17] study as these unavailable in Malaysia. For utility data on CKD Stage 1, dialysis and transplant, these were sourced from published literature from Malaysia (S3 Table). Adverse event-related utility decrements are applied to health state utilities multiplicatively in accordance with ISPOR guidelines [22]. The cost of each event and medication cost were derived from published literature and the Malaysian Ministry of Health (Tables 1 and 2). All costs were inflated to 2022 values based upon the recommendations by Turner and colleagues [23] and presented in the local currency unit as RM, and adjusted to USD to aid in the interpretation, assuming 1USD = RM4.14 [24]. Analysis of cost was conducted from the perspective of the public sector, and other indirect costs were not included.

**Probabilistic and deterministic analyses.** One-way deterministic analysis (DSA), and probabilistic sensitivity analysis (PSA) were conducted to assess the robustness of the analyses. Model parameters were varied within a range of standard error (SE). Variations between ± 10% for probability and cost by ± 20% were applied when there were no specific ranges. A PSA was also carried out to assess the uncertainty of all the parameters simultaneously. We assumed a beta distribution for transitional probability and gamma distribution for cost data. The model

**Table 2. Utility inputs used in this study.**

| Parameter | Mean (SE) | Reference |
|---|---|---|
| **Health state utility** | | |
| CKD 1, >90ml/min per 1.73 m$^2$ | 0.93 (0.005) | Rizal et al. [31] |
| CKD 2, 60–89 ml/min per 1.73 m$^2$ | 0.87 (0.005) | DAPA-CKD trial |
| CKD 3a, 45–59 ml/min per 1.73 m$^2$ | 0.88 (0.002) | DAPA-CKD trial |
| CKD 3b, 30–44 ml/min per 1.73 m$^2$ | 0.88 (0.002) | DAPA-CKD trial |
| CKD 4, 15–29 ml/min per 1.73 m$^2$ | 0.87 (0.003) | DAPA-CKD trial |
| CKD 5 (pre-RRT); <15 ml/min per 1.73 m$^2$ | 0.85 (0.009) | DAPA-CKD trial |
| Dialysis** | 0.85 (0.17) | Surendra et al. [32] |
| Transplant | 0.95 (0.05) | Bavanandan et al. [27] |
| **Adverse event utility decrements** | | |
| Volume depletion | 0.05 (0.01) | DAPA-HF trial [33] |
| Severe hypoglycaemic events | 0.19 (0.00) | Shafie et al. [34] |
| Fractures | 0.072 (0.031) | DAPA-CKD trial |
| Diabetic ketoacidosis | 0.01 (0.01) | DAPA-CKD trial |
| Amputation | 0.266 (0.05) | DAPA-CKD trial |

** Based on haemodialysis

CKD: chronic kidney disease; RRT: renal replacement therapy; SE: standard error

was calculated using 1,000 Monte Carlo simulations and presented in several ways. Firstly, the incremental cost-effectiveness ratios (ICERs) were presented in terms of QALYs and life years. To ease in interpretation, we also presented the results as an incremental net monetary benefit (NMB), where the predicted incremental costs and incremental QALYs is estimated using the Malaysian willingness-to-pay (WTP) threshold of one GDP of RM46,000/QALY [35].

## Results

### Base case analysis

In this study, we predicted that people with CKD treated with dapagliflozin will experience additional health benefits due to an increase in life expectancy and lower rates of CKD progression. Dapagliflozin treatment was estimated to increase life-years (10.01 versus 9.55) and QALYs (8.76 versus 8.35) compared to those on SoC. This is expected to translate to a lower lifetime total cost in those treated with dapagliflozin compared to SoC group (RM 81,814 versus RM 85,464; USD19,762 vs USD20,644), suggesting that the use of dapagliflozin is dominant (cost saving with increased health benefits) in people with CKD (Table 3). The benefits of dapagliflozin were mainly due to the delay in CKD progression, where patients spent a longer time in CKD stage 1 to 3b (6.9 versus 6.0 years) where the benefits are more pronounced.

### Subgroup and sensitivity analyses

Results of the model were robust to changes in assumptions, with all estimates falling below the willingness to pay threshold. The tornado plots show how much the results associated with lower and upper parameter values deviate from the mean values result (Fig 2). Larger cost savings were observed if the model only simulated the effects for the 10 years, when the cost of adverse events were varied, and when the health state utility were varied. Probabilistic analysis found that in 99.9% of simulations were below the RM 46,000/QALY (USD11,111/QALY) gained threshold (Fig 3).

**Table 3. Cost-effectiveness results (per-patient).**

| Outcome | Dapagliflozin + SoC | SoC | Incremental |
|---|---|---|---|
| **Total costs, RM (USD)** | 81,814 (19,762) | 85,464 (20,644) | -3,650 (-882) |
| Drug acquisition, RM (USD) | 3,229 (780) | 442 (107) | 2,787 (673) |
| Disease management cost, RM (USD) | | | |
| Stage 2 | 392 (95) | 384 (93) | 8 (2) |
| Stage 3a | 759 (183) | 690 (167) | 69 (17) |
| Stage 3b | 3,901 (942) | 3,363 (812) | 538 (130) |
| Stage 4 | 4,170 (1007) | 4,148 (1002) | 22 (5) |
| Stage 5 | 1,031 (249) | 1,113 (269) | -83 (-20) |
| Dialysis, RM (USD) | 49,381 (11,928) | 55,123 (13,315) | -5,742 (-1,387) |
| Transplant, RM (USD) | 16,013 (3,868) | 17,496 (4,226) | -1,483 (-358) |
| Adverse events, RM (USD) | 2,898 (700) | 2,680 (648) | 218 (53) |
| **Total life years gained** | 10.01 | 9.55 | 0.46 |
| **Total QALYs gained** | 8.76 | 8.35 | 0.41 |
| **Incremental NMB** | | | 22,528 |
| **ICER (Cost/QALY)**, RM (USD) | | | -RM 8,894 (-2,148) |

ICER: incremental cost-effectiveness ratio; NMB: net monetary benefit; QALY: quality adjusted life year; SoC: standard of care; Assuming 1USD = RM4.14

## Discussion

In this cost effectiveness study, analysis indicates that CKD patients treated with dapagliflozin in addition to SoC would-be a dominant treatment in Malaysia. In particular, the current population will incur a smaller lifetime total cost and slightly gain in life-years and QALYs compared with SoC alone. The analysis showed that there was a saving of RM7,898 (USD1,908) saving for every life year gained and RM8,894 (USD2,148) for every quality adjusted life year gained. Most of these benefits were mainly due to the delay in eGFR decline progression across different disease population. Importantly, the potential benefits associated with delayed CKD progression to dialysis from dapagliflozin can lead to a reduction in the economic burden of CKD treatment in the country. Based upon data from the previous National Health and

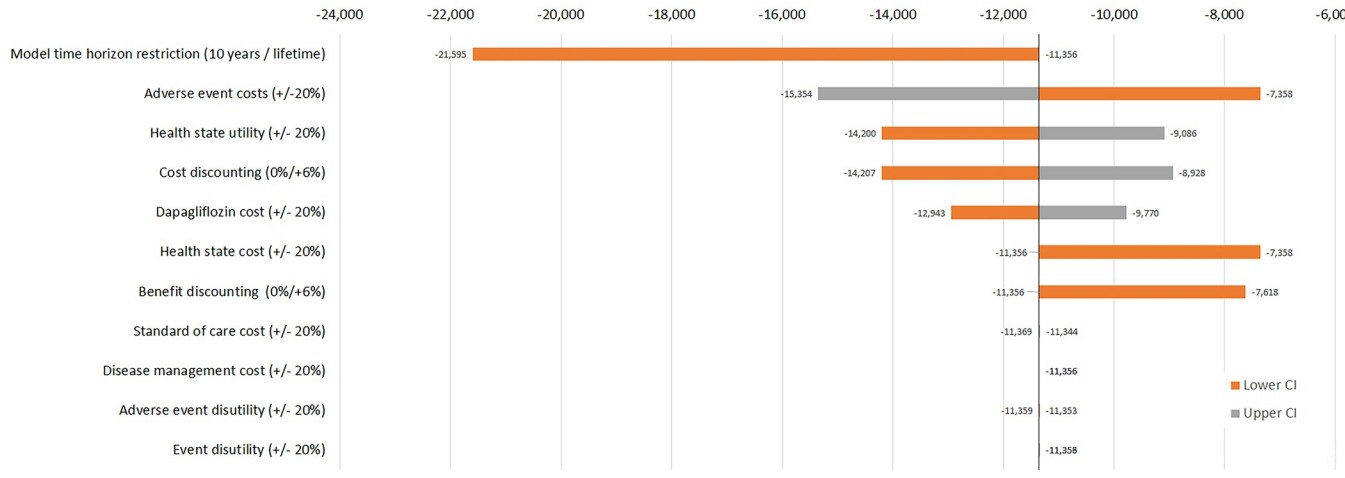

**Fig 2. Changes relative to the base case incremental cost-effectiveness ratio assuming different scenarios or subgroups of interest.**

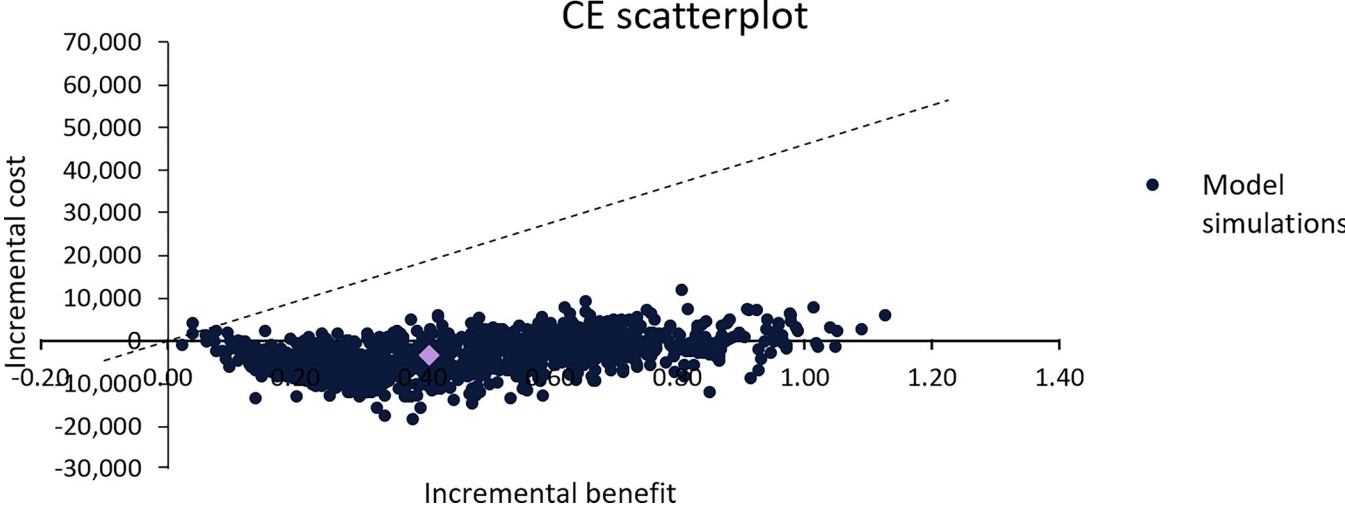

**Fig 3. Cost-effectiveness plane scatterplot.** The established willingness to pay threshold is indicated by the line, and any values below the line are considered dominant.

Morbidity Survey, assuming a prevalence of 15.4% of the Malaysian population have CKD, there will be 3.72 million Malaysians with CKD. As such, the use of dapagliflozin in this population would potentially translate to a savings of RM 13.5 billion (USD 326.2 million) in long term savings if all this population is treated.

The results one-way deterministic analysis also confirmed the cost savings of add-on dapagliflozin in all different scenarios assumed. For example, even when the health state cost (disease management cost) were smaller than those listed, the use of dapagliflozin on top of SoC was still cost-effective. When we simulated the scenario a total of 1,000 times, the results were similar to our base case, showing the robustness of the analyses. Indeed, the results showed that in 99.9% of instances, the use of dapagliflozin in addition to SoC was cost-effective.

Results of this study are in line with published cost-effectiveness study conducted worldwide, including in the UK [36], Thailand [37] and in US [38]. In the study by Varareesangthip in Thailand, the authors similarly found that the add-on dapagliflozin was cost-saving compared to SoC alone in Thailand. The benefit of dapagliflozin was similar to our study where the cohort experienced a delayed CKD progression as it reduces the requirement for dialysis and kidney transplantation, which can offset the costs of dapagliflozin and early CKD treatment.

Nevertheless, there are certain limitations to our model which warrants some discussion. Firstly, the current model does not incorporate the cost savings due to a reduction in heart failure since this information was unavailable in the DAPA-CKD trial. Secondly, as our study population includes mostly people with early stage CKD, it may not reflect the population of those recruited in the DAPA-CKD study. Our analysis assumes that the practice is to treat and start patients early on dapagliflozin treatment where the benefits are more likely to be seen where it can delay disease progression. The analysis represents patients eligible for treatment in Malaysia and thus may not be extrapolated to other countries.

Our model also assumes the broad CKD population, and does not include subgroup analysis such as those with or without diabetes or those with or without heart failure. A higher cost savings is expected especially among those who have diabetes given the greater treatment effects [39, 40]. Importantly, as there is a scarcity of information related to the disease transition matrix in Malaysia, data from the DAPA-CKD was used. The disease transition matrix

and utility values may not reflect the results if the study was implemented in locally. Another limitation was that our model did not include the indirect costs such as productivity loss or transportation costs associated with treatment in our study due to the lack of publicly available data in Malaysia. Finally, our model assumes that the population will stop receiving treatment when they require dialysis in line with current practice. Other potential clinical benefits of using sodium glucose cotransporter-2 inhibitors, such as weight loss, improved glycaemic control and reduction in blood pressure were also not captured in this analysis.

## Conclusion

In summary, we found that the adding dapagliflozin to Soc would be cost saving in people with CKD in Malaysia and demonstrates the potential reduction in kidney events. These findings should be verified in a real-world analysis to help healthcare providers make clinical decision on its use in the future.

## Supporting information

**S1 Checklist. CHEERS 2022 checklist.**
(DOCX)

**S1 Table. CKD transition matrix—Dapagliflozin + SoC—Mean (SE).**
(DOCX)

**S2 Table. CKD transition matrix–Placebo + SoC–Mean (SE).**
(DOCX)

**S3 Table. Modelled baseline characteristics.**
(DOCX)

## Acknowledgments

The authors thank Dr Hooi Lai Seong for her expert reviews, facilitating the team to validate data inputs, as well as providing editorial assistance. Authors acknowledge this analysis was based on the Dapa-CKD trials published.

## Author Contributions

**Data curation:** Shaun Wen Huey Lee.

**Formal analysis:** Soo Kun Lim, Shaun Wen Huey Lee.

**Validation:** Soo Kun Lim, Shaun Wen Huey Lee.

**Writing – original draft:** Soo Kun Lim, Shaun Wen Huey Lee.

**Writing – review & editing:** Soo Kun Lim, Shaun Wen Huey Lee.

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
