## [Decision Letter · Decision Letter 0]

29 Jun 2023

PONE-D-23-16324Cost-effectiveness analysis of dapagliflozin for people with chronic kidney disease in MalaysiaPLOS ONE

Dear Dr. Lee,

Thank you for submitting your manuscript to PLOS ONE. After careful consideration, we feel that it has merit but does not fully meet PLOS ONE’s publication criteria as it currently stands. Therefore, we invite you to submit a revised version of the manuscript that addresses the points raised during the review process.

Please kindly consider the reviewers' comments and make appropriate changes as needed

We look forward to receiving your revised manuscript.

Kind regards,

Yee Gary Ang, MBBS MPH

Academic Editor

PLOS ONE

Journal Requirements:

"This project is supported by AstraZeneca Malaysia."

"The authors thank Dr Hooi Lai Seong for her expert reviews, facilitating the team to validate data inputs, as well as providing editorial assistance. Authors acknowledge this analysis was based on the Dapa-CKD trials published. This study is funded by AstraZeneca Malaysia."

"This project is supported by AstraZeneca Malaysia."

Reviewers' comments:

Reviewer's Responses to Questions

**Comments to the Author**

1. Is the manuscript technically sound, and do the data support the conclusions?

Reviewer #1: Yes

Reviewer #2: Yes

Reviewer #3: Yes

2. Has the statistical analysis been performed appropriately and rigorously? 

Reviewer #1: Yes

Reviewer #2: N/A

Reviewer #3: Yes

3. Have the authors made all data underlying the findings in their manuscript fully available?

Reviewer #1: Yes

Reviewer #2: Yes

Reviewer #3: Yes

4. Is the manuscript presented in an intelligible fashion and written in standard English?

Reviewer #1: Yes

Reviewer #2: Yes

Reviewer #3: Yes

5. Review Comments to the Author

Reviewer #1: Thank you for the opportunity to review the manuscript entitled "Cost-effectiveness analysis of dapagliflozin for people with chronic kidney disease in Malaysia" The study is well conducted and the manuscript well written. I have a few minor comments.

Abstract. It would be preferred if the authors could add under methods that the population in the study was CKD 2-5

Further under results Change to Dapagliflozin in combination with SoC vs SoC alone

And under introduction, A published Markov model .. I suggest that you omit published here.

In the text of the manuscript you frequently refer to DAPA CKD study, please add ref #17 at each of these referrals.

Page 23 add abbreviation SGLT-2 inhibitor which I also believe should be used as a key word.

Please also check the use of spacebar, sometimes used twice and sometimes missing.

For table 1 and 2 change major hypoglycemia to severe hypoglycemia in line with the international classification.

Reviewer #2: Comment

In general, the study follows the standard guidelines of conducting cost-effectiveness studies. Some comments should be addressed before acceptance.

1.This study was conducted from the perspective of the public sector. Why excluded indirect costs from the analysis?

2.What is the unit for adverse event costs? Is it cost per event or cost per admission? How did authors apply these costs in the Markov model? Was it applied only one cycle or all cycles?

3.Why cost of CKD3a was lower than that of CKD2?

4.All costs should be inflated to 2022 instead of 2021.

5.Why did utility of CKD3a, 3b higher than utility of CKD2? In addition, utility of CKD2 is equal to CKD4? Utility of pre-RRT (CKD5) is equal to dialysis? All these data reflect the real situation?

6.How did authors apply utility decrement of adverse events? Did author apply to all cycles or specific cycle? Utility decrement of major hypoglycemic events is quite high (0.19). If this adverse event occurs 5-6 times, the initial utility will become almost zero?

7.Table 3 base-case result: author should display the total cost of each CKD state (if possible). This would help support the conclusion of the study that dapagliflozin arm incurred greater costs of early CKD stage.

8.Tornado diagram: the range of each parameter and axis title should be clearly specified. The longest bar is secondary/tertiary care population only. What is this parameter related to this study? Is it costs, effects, or utility? The third bar from above is adverse event? Which adverse event author referred to? Which health state utility was in the diagram? Explanation of the result should be added in the text.

9.Cohort population (Supplement file). Baseline characteristics of DAPA-CKD trial are quite different from those in this study. For DAPA-CKD, the average age is about 61 years. There is no CKD1 at initial (might be few). The majority of the population are CKD3a (31%), CKD3b (44%). However, this study the starting age is 48.80. The majority of the population are CKD1 (25%), CKD2 (31%). Would these differences have an effect on the findings of this study?

Reviewer #3: Although it is an interesting study , however I have the following comments :

-Treatment related adverse events did not include urinary tract infection and genital infection

- Comparison between diabetic versus non diabetic patients ( it is expected that the annual costs associated with diabetes is high)

-You should mention in the limitations of the study that there is a difference between a clinical outcome in clinical trials and outcomes observed in real world practice

- The results of this cost effective analysis can not be generalized to other countries

6. PLOS authors have the option to publish the peer review history of their article (what does this mean?). If published, this will include your full peer review and any attached files.

Reviewer #1: **Yes: **Johan Jendle

Reviewer #2: No

Reviewer #3: No

---

## [Author Response · Author response to Decision Letter 0]

19 Jul 2023

Yee Gary Ang, MBBS MPH

Academic Editor

PLOS ONE

Journal Requirements:

We take note of the journal’s requirement. The revised manuscript has been revised to the journals requirement including file naming.

"This project is supported by AstraZeneca Malaysia."

We take note. We have now revised the statement to also include the following.

This study is funded by AstraZeneca Malaysia. The funder had no role in study design, data collection and analysis, decision to publish, or preparation of the manuscript

"The authors thank Dr Hooi Lai Seong for her expert reviews, facilitating the team to validate data inputs, as well as providing editorial assistance. Authors acknowledge this analysis was based on the Dapa-CKD trials published. This study is funded by AstraZeneca Malaysia."

"This project is supported by AstraZeneca Malaysia."

We take note. This is now included in our revised cover letter and on the online submission to state the following

This study is funded by AstraZeneca Malaysia. The funder had no role in study design, data collection and analysis, decision to publish, or preparation of the manuscript

We take note. We will not be able to provide the model for other users due to the confidentiality of data but we are happy to provide the information to other people upon reasonable request

We take note and have ensured that the document complies with PLOS One requirement

We take note and have ensured that the document complies with PLOS One requirement

Reviewer #1 had the following comments: 

Comment #1. Thank you for the opportunity to review the manuscript entitled "Cost-effectiveness analysis of dapagliflozin for people with chronic kidney disease in Malaysia" The study is well conducted and the manuscript well written. I have a few minor comments.

We thank the reviewer for the encouraging comments and appreciate the efforts and time taken to review our manuscript. We have addressed these comments as delineated below. 

Comment #2. It would be preferred if the authors could add under methods that the population in the study was CKD 2-5

We thank the reviewer for the suggestion. We have now edited the sentence as follow

A Markov model was adapted to estimate the economic and clinical benefits of dapagliflozin in people with Stage 2 to 5 CKD

Comment #3. Further under results Change to Dapagliflozin in combination with SoC vs SoC alone

We thank the reviewer for the suggestion. The revised sentence now reads as follow:-

In Malaysia, dapagliflozin in combination with SoC was the dominant intervention compared to SoC alone (RM 81,814 versus RM 85,464; USD19,762 vs USD20,644). 

Comment #4. And under introduction, A published Markov model .. I suggest that you omit published here.

We thank the reviewer for the suggestion. We have removed the word published as suggested in our introduction of our abstract. Our revised abstract methods now reads as follow:-

A Markov model was adapted to estimate the economic and clinical benefits of dapagliflozin in people with Stage 2 to 5 CKD

Comment #5. In the text of the manuscript you frequently refer to DAPA CKD study, please add ref #17 at each of these referrals.

We thank the reviewer for the suggestion. We have now included the reference to each of these referrals to clarify this.

Comment #6. Page 23 add abbreviation SGLT-2 inhibitor which I also believe should be used as a key word.

We thank the reviewer for the suggestion. We have included SGLT-2 inhibitor as a keyword as proposed.

Keywords: chronic kidney disease; cost-effectiveness; dapagliflozin; SGLT-2 inhibitor

Comment #7. Please also check the use of spacebar, sometimes used twice and sometimes missing.

We thank the reviewer for the suggestion. We have checked to ensure that these are consistent throughout our manuscript.

Comment #8. For table 1 and 2 change major hypoglycemia to severe hypoglycemia in line with the international classification.

We thank the reviewer for highlighting this. We have now edited the tables and changed major to severe hypoglycaemia as suggested.

Reviewer #2 had the following comments

Comment #1. In general, the study follows the standard guidelines of conducting cost-effectiveness studies. Some comments should be addressed before acceptance.

We thank the reviewer for spending their invaluable time on our manuscript.

Comment #2.This study was conducted from the perspective of the public sector. Why excluded indirect costs from the analysis?

We take note of the suggestion. We only included the direct costs in our model as the indirect costs such as rent, utilities, administrative expenses are currently unavailable in Malaysia. As such, we are unable to account for the indirect cost into our model. We have now included this very important observation in our revised discussion under study limitations. The revised discussion now read as follow:-

Another limitation was that our model did not include the indirect costs such as rentals, administrative costs or manpower costs associated with treatment in our study due to the lack of publicly available data in Malaysia.

Comment #3.What is the unit for adverse event costs? Is it cost per event or cost per admission? How did authors apply these costs in the Markov model? Was it applied only one cycle or all cycles?

We agree this was unclear. The cost for adverse events represents cost per event which may include hospital admission. The cost was derived from published literature whenever possible. Similar to the model developed by Ewen and colleagues, we accounted for events aligned to those related to DAPA-CKD study, where an individual can experience these events multiple times during the cycle. We have now edited the methods section as follow:-

In the current model, only volume depletion, fractures, diabetic ketoacidosis, severe hypoglycaemic events and amputation were accounted for based upon the results of the DAPA-CKD study. The generalised estimating equation using Poisson distribution was used to estimate the recurrent events including volume depletion, fractures, diabetic ketoacidosis, severe hypoglycaemia and amputations. 

Comment #4.Why cost of CKD3a was lower than that of CKD2?

We do note that there appears to be some minor discrepancy in our costs. However, as such costs were not routinely collected in Malaysia, we had to rely on data reported into literature. Nevertheless, these costs data have been validated by our experts who suggest that this may be due to the higher and more aggressive treatment of these individuals. 

Comment #5.All costs should be inflated to 2022 instead of 2021.

We agree. We have revised this to the 2022 value as suggested as the GDP values in 2022 were not available at the time of our study. These have now been revised the results throughout our manuscript based upon these new costs and results.

Comment #6.Why did utility of CKD3a, 3b higher than utility of CKD2? In addition, utility of CKD2 is equal to CKD4? Utility of pre-RRT (CKD5) is equal to dialysis? All these data reflect the real situation?

Yes, we agree that these values may not be ideal. However, as described earlier, information related to utility are rarely collected in a Malaysian population. As such, we have utilised the values related to those collected from the DAPA-CKD study instead. This may not be ideal in this situation but represent the best available evidence. We have similarly requested for two experts to verify this who also agreed with our inputs. This is described in our methods as follow:-

In this analysis, the utility inputs for patients CKD 2-5 were derived from the DAPA-CKD[1] study as these unavailable in Malaysia. For utility data on CKD Stage 1, dialysis and transplant, these were sourced from published literature from Malaysia

Comment #7.How did authors apply utility decrement of adverse events? Did author apply to all cycles or specific cycle? Utility decrement of major hypoglycemic events is quite high (0.19). If this adverse event occurs 5-6 times, the initial utility will become almost zero?

We agree this was unclear. The model utilises a Markov state-transition framework where disease progresses through transition between discrete health state characterised by CKD stage with state specific utility decrement and outcome. As the reviewer correctly pointed out, adverse event is considered a transient event, i.e patient remains in the health state for one cycle and then incur event specific cost and utility decrement. These utility decrement only occurs during that cycle and patient will similarly have a utility value for their respective CKD. 

Comment #8.Table 3 base-case result: author should display the total cost of each CKD state (if possible). This would help support the conclusion of the study that dapagliflozin arm incurred greater costs of early CKD stage.

We thank the reviewer for the excellent suggestion. We have edited the results to conform with the suggestions of the reviewer as follow:-

Comment #9.Tornado diagram: the range of each parameter and axis title should be clearly specified. The longest bar is secondary/tertiary care population only. What is this parameter related to this study? Is it costs, effects, or utility? The third bar from above is adverse event? Which adverse event author referred to? Which health state utility was in the diagram? Explanation of the result should be added in the text.

We agree this was unclear and can lead to confusion. We have now included and added a description of these results to clarify this. Our revised results now read as follow:-

Results of the model were robust to changes in assumptions, with all estimates falling below the willingness to pay threshold. The tornado plots show how much the results associated with lower and upper parameter values deviate from the mean values result (Fig 2). Larger cost savings were observed if the model only simulated the effects for the 10 years, when the cost of adverse events were varied, and when the health state utility were varied. Probabilistic analysis found that in 99.9% of simulations were below the RM 46,000/QALY (USD11,111/QALY) gained threshold (Fig 3). 

Comment #10..Cohort population (Supplement file). Baseline characteristics of DAPA-CKD trial are quite different from those in this study. For DAPA-CKD, the average age is about 61 years. There is no CKD1 at initial (might be few). The majority of the population are CKD3a (31%), CKD3b (44%). However, this study the starting age is 48.80. The majority of the population are CKD1 (25%), CKD2 (31%). Would these differences have an effect on the findings of this study?

We take note of the reviewer’s concern. As highlighted by the reviewer, there are some differences in terms of the population but we believe that the beneficial effects may be applicable to a broader population. We have included these important points in our study discussion under limitations as follow

Nevertheless, there are certain limitations to our model which warrants some discussion. Firstly, the current model does not incorporate the cost savings due to a reduction in heart failure since this information was unavailable in the DAPA-CKD trial. Secondly, our study assumes that the DAPA-CKD population closely mimics the general population in Malaysia. Our analysis assumes that the practice is to treat and start patients early on dapagliflozin treatment where the benefits are more likely to be seen where it can delay disease progression. The analysis represents patients eligible for treatment in Malaysia and thus may not be extrapolated to other countries. 

Our model also assumes the broad CKD population, and does not include subgroup analysis such as those with or without diabetes or those with or without heart failure. A higher cost savings is expected especially among those who have diabetes given the greater treatment effects. As the disease transition matrix and utility values were derived from the DAPA-CKD study, this may not reflect the results if the study was implemented in a real-world setting

Reviewer #3 had the following comments.

Comment #1. Although it is an interesting study , however I have the following comments : -Treatment related adverse events did not include urinary tract infection and genital infection

We agree this is a common and important adverse event for people taking SGLT-2 inhibitors. However, given that these events are considered transient and does not incur significant cost to the healthcare system, we did not account for these in our model. Only Grade 3-5 adverse events were included in our model. To clarify this, we have included this in our revised methods as follow:-

Dapagliflozin, like many other SGLT-2 inhibitors has a well-established safety profile. As such, in the current model, only grades 3 to 5 adverse events that have been found to be significantly different in the DAPA-CKD study[1] were accounted for, including volume depletion, fractures, diabetic ketoacidosis, severe hypoglycaemic events and amputation.

Comment #2 Comparison between diabetic versus non diabetic patients ( it is expected that the annual costs associated with diabetes is high)

We thank the reviewer for the suggestion. However, we were unable to account for these due to the limited information that were available from the DAPA-CKD study which did not report this very important observation. To clarify this, we have included this in the discussion under study limitations as follow:-

Our model also assumes the broad CKD population, and does not include subgroup analysis such as those with or without diabetes or those with or without heart failure. A higher cost savings is expected especially among those who have diabetes given the greater treatment effects.

Comment #3 You should mention in the limitations of the study that there is a difference between a clinical outcome in clinical trials and outcomes observed in real world practice

We agree and thank the reviewer for this very important suggestion. We have included this in our revised discussion section as follow:-

As the disease transition matrix and utility values were derived from the DAPA-CKD study, this may not reflect the results if the study was implemented in a real-world setting.

Comment#4 The results of this cost effective analysis can not be generalized to other countries

We agree. We have revised the discussion section as requested. The revised discussion limitation are as follow

The analysis represents patients eligible for treatment in Malaysia and thus may not be extrapolated to other countries.

---

## [Decision Letter · Decision Letter 1]

31 Jul 2023

PONE-D-23-16324R1Cost-effectiveness analysis of dapagliflozin for people with chronic kidney disease in MalaysiaPLOS ONE

Dear Dr. Lee,

Thank you for submitting your manuscript to PLOS ONE. After careful consideration, we feel that it has merit but does not fully meet PLOS ONE’s publication criteria as it currently stands. Therefore, we invite you to submit a revised version of the manuscript that addresses the points raised during the review process.

We have sent the manuscript to 2 reviewers.

Please make the changes as indicated. 

We look forward to receiving your revised manuscript.

Kind regards,

Yee Gary Ang, MBBS MPH

Academic Editor

PLOS ONE

Journal Requirements:

Reviewers' comments:

Reviewer's Responses to Questions

**Comments to the Author**

1. If the authors have adequately addressed your comments raised in a previous round of review and you feel that this manuscript is now acceptable for publication, you may indicate that here to bypass the “Comments to the Author” section, enter your conflict of interest statement in the “Confidential to Editor” section, and submit your "Accept" recommendation.

Reviewer #1: All comments have been addressed

Reviewer #2: (No Response)

2. Is the manuscript technically sound, and do the data support the conclusions?

Reviewer #1: Yes

Reviewer #2: Yes

3. Has the statistical analysis been performed appropriately and rigorously? 

Reviewer #1: Yes

Reviewer #2: N/A

4. Have the authors made all data underlying the findings in their manuscript fully available?

Reviewer #1: Yes

Reviewer #2: Yes

5. Is the manuscript presented in an intelligible fashion and written in standard English?

Reviewer #1: Yes

Reviewer #2: Yes

6. Review Comments to the Author

Reviewer #1: The authors have sucessfully responded to the comments and the manuscript is now suitable for publication.

Reviewer #2: Overall, authors have addressed almost all comments. However, some issues need to be clarified. Authors should use track change in the text for the revised version.

The baseline characteristics of cohort population in DAPA-CKD trial are quite different from those in this study. The majority of cohort population in this study is at the early stage of CKD. The term “closely mimic” in the second limitation might not be appropriate.

Another point is indirect cost. In general, indirect costs are related to productivity loss or time loss. Manpower costs or administrative costs are considered as direct cost in economic analysis.

7. PLOS authors have the option to publish the peer review history of their article (what does this mean?). If published, this will include your full peer review and any attached files.

Reviewer #1: **Yes: **Johan Jendle

Reviewer #2: No

---

## [Author Response · Author response to Decision Letter 1]

1 Aug 2023

Yee Gary Ang, MBBS MPH

Academic Editor

PLOS ONE

Dear Prof Ang,

Thank you for kindly considering our article. Based upon the comments from both reviewers, we have made the changes as requested as follow:-

Reviewer #1 had the following comment. The authors have successfully responded to the comments and the manuscript is now suitable for publication.

Thank you 

Reviewer #2 had the following comments.

Comment # 1. Overall, authors have addressed almost all comments. However, some issues need to be clarified. Authors should use track change in the text for the revised version.

We take note. The submission includes a track changes as per journal requirement.

Comment #2. The baseline characteristics of cohort population in DAPA-CKD trial are quite different from those in this study. The majority of cohort population in this study is at the early stage of CKD. The term “closely mimic” in the second limitation might not be appropriate.

We agree. We have now revised the sentence to read as follow:-

Secondly, as our study population includes mostly people with early stage CKD, it may not reflect the population of those recruited in the DAPA-CKD study.

Comment #3. Another point is indirect cost. In general, indirect costs are related to productivity loss or time loss. Manpower costs or administrative costs are considered as direct cost in economic analysis.

We agree. We have now revised the sentence to read as follow

Another limitation was that our model did not include the indirect costs such as productivity loss or transportation costs associated with treatment in our study due to the lack of publicly available data in Malaysia.

We trust that these changes will meet the requirements of the reviewer.

Yours truly

Shaun Lee

On behalf of the authors

---

## [Decision Letter · Decision Letter 2]

10 Nov 2023

PONE-D-23-16324R2Cost-effectiveness analysis of dapagliflozin for people with chronic kidney disease in MalaysiaPLOS ONE

Dear Dr. Lee,

Thank you for submitting your manuscript to PLOS ONE. After careful consideration, we feel that it has merit but does not fully meet PLOS ONE’s publication criteria as it currently stands. Therefore, we invite you to submit a revised version of the manuscript that addresses the points raised during the review process.

We invited 4 reviewers with mixed comments.

Please address the remaining comments

We look forward to receiving your revised manuscript.

Kind regards,

Yee Gary Ang, MBBS MPH

Academic Editor

PLOS ONE

Journal Requirements:

Reviewers' comments:

Reviewer's Responses to Questions

**Comments to the Author**

1. If the authors have adequately addressed your comments raised in a previous round of review and you feel that this manuscript is now acceptable for publication, you may indicate that here to bypass the “Comments to the Author” section, enter your conflict of interest statement in the “Confidential to Editor” section, and submit your "Accept" recommendation.

Reviewer #1: (No Response)

Reviewer #2: All comments have been addressed

Reviewer #4: (No Response)

Reviewer #5: All comments have been addressed

2. Is the manuscript technically sound, and do the data support the conclusions?

Reviewer #1: Partly

Reviewer #2: Yes

Reviewer #4: Yes

Reviewer #5: Yes

3. Has the statistical analysis been performed appropriately and rigorously? 

Reviewer #1: Yes

Reviewer #2: N/A

Reviewer #4: Yes

Reviewer #5: Yes

4. Have the authors made all data underlying the findings in their manuscript fully available?

Reviewer #1: Yes

Reviewer #2: Yes

Reviewer #4: Yes

Reviewer #5: Yes

5. Is the manuscript presented in an intelligible fashion and written in standard English?

Reviewer #1: Yes

Reviewer #2: Yes

Reviewer #4: No

Reviewer #5: Yes

6. Review Comments to the Author

Reviewer #1: (No Response)

Reviewer #2: There is no further comment. All comments have been addressed. The findings of the study are beneficial to healthcare system in Malaysia.

Reviewer #4: In the present manuscript the authors aimed to evaluate the cost-effectiveness of dapagliflozin in Malaysian population with CKD. Please find my minor concerns:

1. The authors stated that “Disease progression was captured using parameters based upon the DAPA-CKD trial” Is there any local study to assure this approach?

2. I really suggest to revise the content of the Introduction. There are numerous repetition, for instance: “Complications associated with CKD include increased risk of all-cause and cardiovascular mortality, kidney disease progression, and fractures” and, “CKD is also associated with an increase in cardiovascular diseases and overall mortality”. OR “has substantial societal and economic impact” and “In addition, it is also associated health-related quality of life (HRQoL) impairments and increase in healthcare costs”. Also, the English needs revision (e.g., “to improved glycaemic control.”

3. I did not understand why the cost of CKD grade 2 is higher compared to CKD 3a (Table 1)?

4. Please explain in Table 1, what means volume depletion in the current study

Reviewer #5: I would like to express my sincere gratitude to you for taking the time and effort to address all the comments and feedback provided on your manuscript. Your dedication to refining your work and incorporating the suggested improvements is greatly appreciated.

I want to wish you the best of luck as you move forward with the publication process. I have no doubt that your hard work and commitment to your research will lead to a successful and impactful publication.

7. PLOS authors have the option to publish the peer review history of their article (what does this mean?). If published, this will include your full peer review and any attached files.

Reviewer #1: **Yes: **Johan Jendle

Reviewer #2: No

Reviewer #4: No

Reviewer #5: No

---

## [Author Response · Author response to Decision Letter 2]

26 Nov 2023

Yee Gary Ang, MBBS MPH

Academic Editor

PLOS ONE

Dear Prof Ang,

Thank you for kindly considering our article. Based upon the comments from both reviewers, we have made the changes as requested as follow:-

Reviewer #4 had the following comment

Comment #1. The authors stated that “Disease progression was captured using parameters based upon the DAPA-CKD trial” Is there any local study to assure this approach?

We take note of the reviewer’s comment. In our model, we used the benefits of delay in disease progression using dapagliflozin based upon the data from DAPA-CKD as this information was unavailable locally. Currently, dapagliflozin is only available in Malaysia as a restricted access for people with diabetes at a very limited number. However, disease progression for people with chronic kidney disease was based upon the data from Malaysia’s renal registry. This is described in the methods section as follow.

Disease progression and mortality

Disease progression was captured using parameters based upon the DAPA-CKD trial[17] data supplemented with locally relevant information where applicable[21]. 

We have also edited the discussion section as a study limitation as follow:-

Importantly, as there is a scarcity of information related to the disease transition matrix in Malaysia, data from the DAPA-CKD was used. The disease transition matrix and utility values may not reflect the results if the study was implemented in locally.

Comment #2. I really suggest to revise the content of the Introduction. There are numerous repetition, for instance: “Complications associated with CKD include increased risk of all-cause and cardiovascular mortality, kidney disease progression, and fractures” and, “CKD is also associated with an increase in cardiovascular diseases and overall mortality”. OR “has substantial societal and economic impact” and “In addition, it is also associated health-related quality of life (HRQoL) impairments and increase in healthcare costs”. Also, the English needs revision (e.g., “to improved glycaemic control.”

We take note and thank the reviewer for pointing this out. We have now revised the content of the introduction to be more succinct as follow:-

Chronic kidney disease (CKD) is estimated to affect 8–16% people globally[1, 2]. In people with CKD, they are associated with an increased risk of all-cause and cardiovascular mortality and fractures.[3-5] This has a far reaching impact, and is associated health-related quality of life (HRQoL) impairments, but also has substantial societal and economic impact.[6] As such, there is significant benefits from halting or delaying CKD progression in the general population.[7-9] 

In Malaysia, the prevalence of CKD has been increasing over the past decade. The National Health and Morbidity Survey (NHMS) in 2011 reported that the prevalence of CKD was 9.07%.[10] This prevalence increased to 15.48% in the recent prevalence study in 2018.[11] Despite the widespread problem of CKD, treatment options are limited; with only angiotensin-converting enzyme inhibitors (ACEi) and angiotensin receptor blockers (ARB) being the only medications that have consistently shown to slow disease progression.[12-14] Recently, sodium-glucose co-transporter-2 (SGLT2) inhibitor were reported to have cardiovascular and renal benefits in addition to improved glycaemic control.[15, 16]

Comment #3. I did not understand why the cost of CKD grade 2 is higher compared to CKD 3a (Table 1)?

We agree. This information may be confusing but the information was derived from our literature review from a study by Azmi and colleagues which reported a higher cost for CKD grade 2. One plausible reason could be the costing approach used but these were not significantly different between both groups and was considered minor. Nevertheless, we have validated these values with our experts who agreed these are within the usual range of RM 400-500 as per practice in Malaysia. We hope this will help clarify the doubts of the reviewer.

Comment #4. Please explain in Table 1, what means volume depletion in the current study

We agree this was unclear. We used the definition of the DAPA-CKD study which stated the following, Volume depletion was defined as an event that resulted in loss of fluid such as dehydration, hypovolemia, or hypotension requiring hospitalisation. To clarify this, we have included the definition as a footnote in Table 1.

Reviewer #5: I would like to express my sincere gratitude to you for taking the time and effort to address all the comments and feedback provided on your manuscript. Your dedication to refining your work and incorporating the suggested improvements is greatly appreciated. I want to wish you the best of luck as you move forward with the publication process. I have no doubt that your hard work and commitment to your research will lead to a successful and impactful publication.

We wish to thank the reviewer for the encouraging comments.

We trust that these changes will meet the requirements of the reviewer.

Yours truly

Shaun Lee

On behalf of the authors

---

## [Decision Letter · Decision Letter 3]

6 Dec 2023

Cost-effectiveness analysis of dapagliflozin for people with chronic kidney disease in Malaysia

PONE-D-23-16324R3

Dear Dr. Lee,

We’re pleased to inform you that your manuscript has been judged scientifically suitable for publication and will be formally accepted for publication once it meets all outstanding technical requirements.

Kind regards,

Yee Gary Ang, MBBS MPH

Academic Editor

PLOS ONE

Additional Editor Comments (optional):

Reviewers' comments:

Reviewer's Responses to Questions

**Comments to the Author**

1. If the authors have adequately addressed your comments raised in a previous round of review and you feel that this manuscript is now acceptable for publication, you may indicate that here to bypass the “Comments to the Author” section, enter your conflict of interest statement in the “Confidential to Editor” section, and submit your "Accept" recommendation.

Reviewer #4: All comments have been addressed

2. Is the manuscript technically sound, and do the data support the conclusions?

Reviewer #4: Yes

3. Has the statistical analysis been performed appropriately and rigorously? 

Reviewer #4: Yes

4. Have the authors made all data underlying the findings in their manuscript fully available?

Reviewer #4: Yes

5. Is the manuscript presented in an intelligible fashion and written in standard English?

Reviewer #4: Yes

6. Review Comments to the Author

Reviewer #4: All the comments have been addressed. The article has been revised English-wise and other methodological issues have been resolved.

7. PLOS authors have the option to publish the peer review history of their article (what does this mean?). If published, this will include your full peer review and any attached files.

Reviewer #4: **Yes: **PARHAM SADEGHIPOUR

---

## [Editor Report · Acceptance letter]

12 Dec 2023

PONE-D-23-16324R3 

PLOS ONE

Dear Dr. Lee, 

I'm pleased to inform you that your manuscript has been deemed suitable for publication in PLOS ONE. Congratulations! Your manuscript is now being handed over to our production team.

Kind regards, 

on behalf of

Dr. Yee Gary Ang 

Academic Editor

PLOS ONE